# Effect of X-ray free-electron laser-induced shockwaves on haemoglobin microcrystals delivered in a liquid jet

Marie Luise Grünbein[1], Alexander Gorel[1], Lutz Foucar[1], Sergio Carbajo[2], William Colocho [2], Sasha Gilevich[2], Elisabeth Hartmann[1], Mario Hilpert [1], Mark Hunter [2], Marco Kloos[1,4], Jason E. Koglin[2,5], Thomas J. Lane[2,6], Jim Lewandowski[2], Alberto Lutman [2], Karol Nass [1,7], Gabriela Nass Kovacs[1], Christopher M. Roome[1], John Sheppard[2], Robert L. Shoeman[1], Miriam Stricker [1,8], Tim van Driel[2], Sharon Vetter[2], R. Bruce Doak[1], Sébastien Boutet [2], Andrew Aquila [2], Franz Josef Decker[2], Thomas R. M. Barends[1], Claudiu Andrei Stan [3✉] & Ilme Schlichting [1✉]

X-ray free-electron lasers (XFELs) enable obtaining novel insights in structural biology. The recently available MHz repetition rate XFELs allow full data sets to be collected in shorter time and can also decrease sample consumption. However, the microsecond spacing of MHz XFEL pulses raises new challenges, including possible sample damage induced by shock waves that are launched by preceding pulses in the sample-carrying jet. We explored this matter with an X-ray-pump/X-ray-probe experiment employing haemoglobin microcrystals transported via a liquid jet into the XFEL beam. Diffraction data were collected using a shock-wave-free single-pulse scheme as well as the dual-pulse pump-probe scheme. The latter, relative to the former, reveals significant degradation of crystal hit rate, diffraction resolution and data quality. Crystal structures extracted from the two data sets also differ. Since our pump-probe attributes were chosen to emulate EuXFEL operation at its 4.5 MHz maximum pulse rate, this prompts concern about such data collection.

[1] Max Planck Institute for Medical Research, Jahnstrasse 29, Heidelberg, Germany. [2] SLAC National Accelerator Laboratory, Menlo Park, CA, USA. [3] Department of Physics, Rutgers University Newark, Newark, NJ, USA. [4] Present address: European XFEL GmbH, Schenefeld, Germany. [5] Present address: Los Alamos National Laboratory, Los Alamos, NM, USA. [6] Present address: Center for Free-Electron Laser Science, DESY, Hamburg, Germany. [7] Present address: Paul Scherrer Institut, Villigen, Switzerland. [8] Present address: Department of Statistics, University of Oxford, Oxford, UK. ✉email: claudiu.stan@rutgers.edu; ilme.schlichting@mpimf-heidelberg.mpg.de

X-ray free-electron lasers (XFELs) are the brightest X-ray sources currently available and so well suited to investigating weakly scattering objects, such as nanocrystals[1,2] and single noncrystalline particles, such as viruses[3]. Moreover the femtosecond duration of XFEL pulses matches chemical timescales, allowing the dynamics of matter to be studied in a time-resolved manner[4–6] and enabling characterization of highly radiation-sensitive objects[7–9]. However, the undisputed benefits of XFELs for structural biology are tied to unique experimental challenges: upon forming a diffraction image, the XFEL pulse annihilates its target, a process dubbed "diffraction before destruction"[10]. XFEL diffraction data must hence be collected serially (serial femtosecond crystallography, or SFX) for which rates of sample replenishment, detector readout and X-ray pulse arrival must be compatible. The first generation of non-superconducting XFELs delivers X-ray pulses at 10–120 Hz. Proven techniques of sample replenishment at these rates include delivery in either low[11] or high[12,13] viscosity free-stream microjets, and presentation of sample via rapidly translatable fixed mounts[14–16]. Low (aqueous)-viscosity microjets are typically only a few microns in diameter and consequently they generate very little X-ray background scattering. This is highly advantageous, but the small jet diameter is inexorably tied to high jet speed. At first-generation XFEL pulse rates (120 Hz and below), most of the sample (>99%) in such jets flows past the scattering point in between XFEL pulses, unprobed and therefore wasted. At MHz pulse rates, in contrast, the jet displacement between XFEL pulses just suffices to flush the damaged jet section downstream. Little or no sample is wasted. For this reason, but also to satisfy increasing demands on XFEL beam time, high-repetition rate XFELs have been awaited eagerly.

The European XFEL (EuXFEL) in Germany is the first XFEL to operate at MHz repetition rates[17]. Designed to provide up to 27,000 pulses per second (delivered in ten pulse trains per second at a 4.5 MHz repetition rate within each train), this increases the number of pulses per second by a factor of 225 or more compared to previous XFELs. To exploit this increase, rapid sample delivery is essential and exactly this is provided by low viscosity, small diameter, high-speed liquid microjets[18–21] as produced by a gas dynamic virtual nozzle (GDVN)[11]. However, the intense XFEL pulse that makes SFX possible also isochorically raises the energy density abruptly and enormously within a microscopic portion of the jet. As shown in a publication by Stan et al.[22], this leads to complete vaporization of a jet segment at the point of irradiation. Shockwaves are launched, propagating supersonically along the jet stream and outrunning the explosive gap that opens up behind them. Associated with the shock front is a nanosecond-duration pressure jump of roughly 0.1–1 GPa (1–10 kbar), which suffices to induce protein unfolding[23]. Moreover, the upstream shock necessarily intercepts sample being carried downstream towards the XFEL scattering point. Accordingly, there are two critical issues for MHz data collection using liquid jets: (i) the formation of the jet gap precludes subsequent XFEL measurements until a contiguous jet has been re-established at the interaction point. Since the gap is finite in extent and eventually flushes downstream at the jet speed, the jet will always heal in time for the next exposure if the jet is fast enough. (ii) The shock wave propagating upstream along the jet may damage or change[24] the sample species carried by the jet (microcrystals in case of SFX; molecules in case of small-angle X-ray scattering (SAXS) or spectroscopy). The spatiotemporal extent of these processes and the severity of the damage they induce in samples can limit the maximum rate at which unblemished serial XFEL data can be collected. Understanding of these issues is critical for establishing the maximally usable repetition rates at MHz rate XFELs.

Protein structures often contain cavities for ligands, cofactors or water molecules. These, as well as loosely folded loops or water-mediated structural interactions, are important for the structural flexibility intrinsic to protein functionality. Such "packing defects" and also the large solvent channels in protein crystals render proteins and their crystals sensitive to applied pressure[23,25–27]. The identification of potential ligand binding pockets in protein structures is often an important step in rational drug design, with the size and shape of cavities determining putative binding site volumes[28]. XFELs have been advocated to accelerate structure-based drug discovery[29]. Clearly, XFEL shock-induced structural changes may complicate this task and thwart valid structural interpretation.

The first SFX experiments at EuXFEL demonstrated that SFX data collection is possible at 1.1 MHz repetition rate. Specifically, no significant differences were observed for diffraction data collected using the first X-ray pulse in a pulse train compared to that of subsequent pulses[19,20,30]. These experiments were conducted during the early stages of accelerator operation; the EuXFEL design specification of 4.5 MHz repetition rate[17] was not available. Thus it has not yet been established whether diffraction data collected at 4.5 MHz is unaffected by shockwaves.

Here, we describe a time-resolved X-ray pump X-ray probe SFX experiment, performed at the Linac Coherent Light Source (LCLS) in Menlo Park, USA, to test the effect of a shockwave launched by the first (pump) X-ray pulse on microcrystals further upstream in a liquid microjet, as probed by the second (probe) X-ray pulse at the upstream point after a 122.5 ns delay. In addition to collecting X-ray pump X-ray probe SFX data of haemoglobin microcrystals, we also collected diffraction data using a single pulse, in which case shockwave damage is necessarily precluded. We compare the quality of the diffraction data, as well as the crystal structures determined using the single-pulse and pump–probe data, respectively. Our findings have implications for the anticipated 4.5 MHz repetition rate of EuXFEL.

## Results and discussion

**Experimental design.** The design repetition rate of EuXFEL is 4.5 MHz within each pulse train, corresponding to a pulse-to-pulse separation of 222 ns (ref. [17]). Temporal XFEL pulse separations of this magnitude are available at the LCLS using a "two-bunch" mode to produce closely spaced pulse pairs, in which two pulses can be separated from 0.35 ns up to hundreds of nanoseconds[31]. These pulse pairs are delivered at the usual 120 Hz repetition rate of the LCLS. In addition, the pulses can be offset spatially along a sample-carrying liquid microjet. Pulse-pair operation comes at the cost of higher instability in XFEL photon and pulse energies, but is ideally suited to explore whether one XFEL pulse can influence the diffraction quality or observed structure of protein microcrystals, as subsequently probed by a following pulse. The pulse-pair spacing and offset can be set to directly emulate the anticipated 4.5 MHz operation of EuXFEL.

We performed an X-ray pump/X-ray probe experiment (LR76 February 2018) at the LCLS in the microfocus chamber of the coherent X-ray imaging (CXI) end station[32]. Using GDVN injection, haemoglobin microcrystals were directed into the X-ray interaction region within a liquid microjet (diameter ~5 μm, speed ~50 m s$^{-1}$). The jet was first exposed to the pump pulse and then, after a chosen time delay $\Delta t$, to the probe pulse. To ensure that the probe pulse fully avoided the explosive gap induced by the pump pulse, the probe beam was vertically displaced $\Delta x = 5$ μm upstream along the jet axis (towards the GDVN nozzle; Fig. 1). Since a microcrystal-carrying liquid jet can alter its position and shape significantly even on 100 ns timescales, it was important to experimentally monitor and

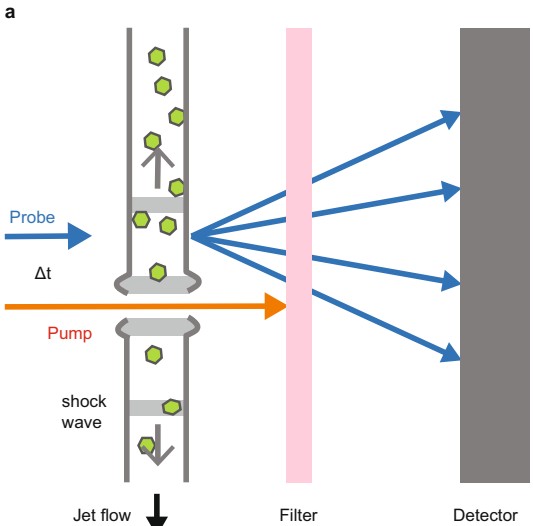

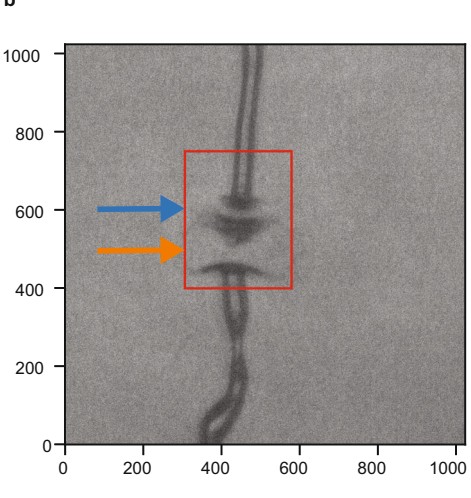

**Fig. 1 Experimental setup. a** Haemoglobin microcrystals were injected into the XFEL beam using a gas dynamic virtual nozzle injector[11]. The first ~30 fs X-ray pulse (photon energy above the iron K-edge (7.112 keV) and a pulse energy of up to ~0.1 mJ, average value 0.03 mJ) was used as a pump, isochorically heating the jet and launching a shockwave (depicted in grey) that propagates upstream and downstream of the interaction region. The shock wave propagates at supersonic speeds and outruns the explosive gap formed in the jet. The scattered X-rays were absorbed by a thin iron filter and did not reach the detector. After 122.5 ns a sample segment upstream of the pump pulse was hit by a second ~30 fs X-ray pulse (photon energy just below the iron K-edge and a pulse energy of ~0.9 mJ), which was displaced by ~5 μm towards the nozzle. In this case, the scattered X-rays passed through the iron filter, reaching the detector. The setup differs from previous two-colour X-ray pump/X-ray probe experiments[33,34] due to the displacement of the pump pulse. **b** Femtosecond snapshot image of the jet a few nanoseconds after the probe pulse had interacted with the jet. Explosions induced by the pump (orange arrow) and probe (blue arrow) pulse are clearly visible. Although present in the jet, the shockwaves are not visible in the camera images. In jets this small, the optical path difference induced by shock compression does not suffice to observe the shock. A movie showing consecutive pump–probe pairs hitting the jet is available as Supplementary Movie 1. The jet instabilities (wiggling) are ascribed to the presence of crystals in the jet. These instabilities are largely absent in homogeneous (crystal-free) jets and become more pronounced with increasing crystal concentration.

exclude such instances. We therefore characterized the liquid jet shortly after interaction with the probe pulse by use of femtosecond snapshot imaging to identify those shots, in which (i) the pump pulse hit the jet and (ii) the jet shape was such as to support shockwave propagation upstream to the probe position (Supplementary Methods, and Supplementary Figs. 1 and 2). Only these diffraction images were then used for subsequent data reduction and analysis. To separate the diffraction patterns of the pump and probe X-ray pulses, their photon energies were set respectively to ~40 eV above and below the iron K-absorption edge (~7.11 keV). A thin iron foil placed in front of the CSPAD detector then transmitted the probe pulse, but blocked the pump pulse[33,34] (Fig. 1). The pulse energy of the combined pump and probe pulses (each ~ 30 fs long) was ~0.9 mJ at the XFEL source.

To emulate EuXFEL operation at 4.5 MHz, the spatial offset of our probe beam from the pump beam must be taken into account in choosing the delay time $\Delta t$. Since a shockwave-induced pressure jump decays as a function of distance travelled[35], the germane quantity is the distance a GDVN jet travels between 4.5 MHz pulses, which depends on the jet speed. Assuming that jets of at least 50 m s$^{-1}$ are needed to flush the explosive gap at 4.5 MHz operation, the required time delay is 122.5 ns. With this jet speed and delay, our 5 μm offset pump/probe measurements correspond to 4.5 MHz EuXFEL operation with a 50 m s$^{-1}$ jet probed by the non-offset EuXFEL beam (Supplementary Note 1). We used carbonmonoxy haemoglobin (Hb.CO) as a model system to explore potential shock effects: the crystal form used here (44% solvent) displays a high degree of plasticity, and thereby an ability to accommodate changes in its quaternary structure or unit cell constants. In view of the sensitivity of our Hb.CO crystals to environmental changes, we collected not just pump/probe data but also, by suppressing the pump pulse, a single-pulse reference data set.

**Effect of the X-ray pump pulse on haemoglobin SFX data**. We acquired 342,609 and 138,453 detector images for the pump–probe and single-pulse setups, from which we identified 43,003 and 25,742 crystal hits (defined as ≥10 diffraction spots/image), respectively. Thus, the average hit rate dropped from 19% in the single-pulse reference data set to 13% in the pump–probe data set (Fig. 2a). Since the hit rate was calculated prior to the filtering analysis, it includes hits in which jet shape does not support shock wave propagation. A valid concern is therefore that the drop in hit rate might be caused by poor beam-jet alignment, either by shooting into the jet gap or by otherwise missing the jet. However, the experiment was designed to observe two distinct gaps (Fig. 1 and Supplementary Movie 1), which generally rules out making gaps that are too large (see also ref. [36]). Nevertheless, explosions of irregularly shaped jets are more complicated and cannot be predicted based on what we know about XFEL explosions. Thus, it cannot be excluded that the decrease in crystal hit rate at higher pump pulse energies is not solely due to a decrease in crystal quality, an additional factor may be the increasing jet disruption. While it is possible that the X-ray fluence differs between the two data collection modes, the pointing did not (Supplementary Fig. 3). The difference in hit rates suggests that 6% of the crystals in the pump–probe data were damaged to such an extent that their diffraction patterns no longer qualified as hits (i.e., <10 diffraction spots per image).

A total of 14,434 (pump–probe) and 24,083 (single-pulse) crystal hits passed the filtering analysis steps (see "Methods" section), which ensures that characterization of pulse energies was reliable and that pump–probe hits were indeed exposed to a shock wave. The indexing rates (total number of indexed diffraction images) of the filtered data were 24% (3531 indexed images after filtering) and 23% (5541 indexed images after filtering) for the pump–probe and single-pulse data sets,

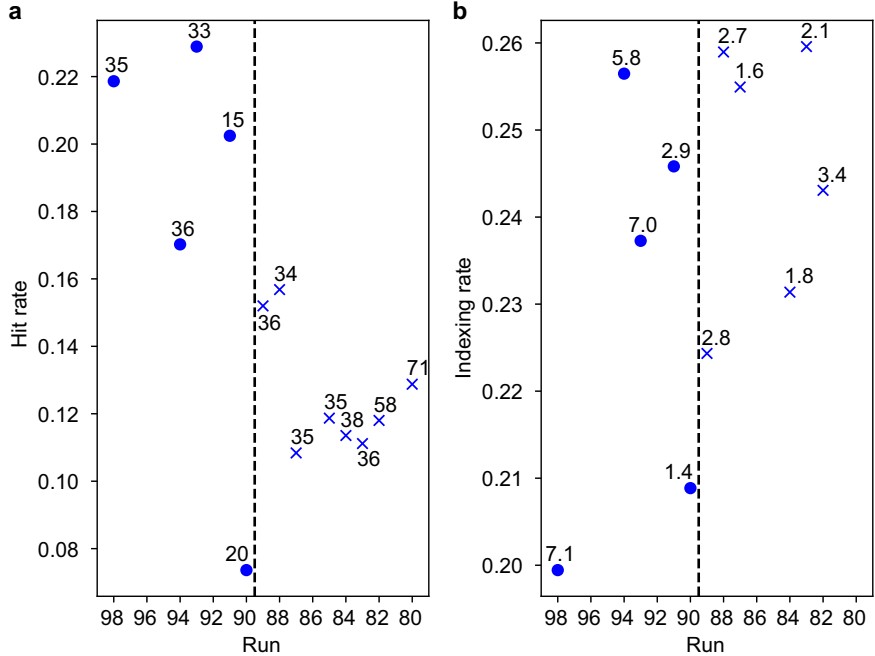

**Fig. 2 Hit and indexing rates in pump–probe and probe-only runs. a** Hit rate as a function of run number for single-pulse (circles) and pump–probe (crosses) runs. The hit rate was calculated as the number of hits divided by the number of shots comprising a run. The average value of the hit rate is 13% in the pump–probe data set and 19% in the single-pulse data set. All hits prior to filtering were taken into account (43,003 hits for the pump–probe and 25,742 hits for the single-pulse data set). The number above each data point indicates the number (in thousands) of X-ray shots per run. **b** Indexing rate as a function of run number for single-pulse (circles) and pump–probe (crosses) runs. The indexing rate was calculated as the number of indexed hits divided by the total number of hits in a run. The average indexing rate was 24% in the pump–probe data set and 23% in the probe-only data set, leading to 3531 and 5541 indexed images in the pump–probe and single-pulse data set, respectively. Only hits satisfying the diode signal and jet image filtering conditions (see "Methods" section) were taken into account. The number above each data point indicates the number (in thousands) of indexed hits per run. For two runs (run 80 and 85), the jet imaging time delay varied and the jet images could thus not be analysed as to whether a shock wave had been launched that could successfully propagate to the jet segment to be probed. Therefore, all hits from these runs were excluded from the analysis and the post-filtering indexing rate is not defined.

respectively (Fig. 2b). Determination of the unit cell constants for the pump–probe data was complicated by the fact that only the average photon energy of each pulse pair could be measured (see Supplementary Note 2). We therefore adopted the most conservative assumption, namely that the unit cell lengths were not affected by the pump pulse, and used the same unit cell dimensions for the single-pulse and pump–probe data. Observed structural changes are then solely due to differences in the diffraction intensity modulations and not to modified sampling of reciprocal space.

The high-resolution limit of strong Bragg spots (signal-to-noise ratio $I/\sigma(I) \geq 4$) of our haemoglobin pump–probe data dropped on average by ~0.3 Å compared to the single-pulse data (Fig. 3a). This resolution drop on individual images is consistent across comparable probe pulse energies (Fig. 3b), and thus not caused by the slightly different probe pulse energy distributions (Supplementary Figs. 4 and 5). Consistent with the decrease in resolution in the individual images, the integrated diffraction intensities also show a clear drop in data quality metrics vs. resolution ($R_{split}$, $I/\sigma(I)$, $CC_{1/2}$), and consequently a reduction in the overall resolution of the data set (Table 1 and Supplementary Fig. 6). These results imply a significant degradation in diffraction from haemoglobin microcrystals in the pump–probe data set. Since radiation damage inflicted by the pump pulse can be excluded as an underlying cause (see Supplementary Note 3, Supplementary Figs. 7–11 and Supplementary Software 1), degradation due to the shock wave is the most likely explanation. The magnitude of the shock wave depends on pump pulse energy and so we would expect to observe increased damage at higher pump pulse energy.

We do observe an enhanced degradation in data quality, specifically a decrease in resolution, as a function of pump pulse energy (Fig. 3). However, the energies of pump and probe pulses are strongly anti-correlated, resulting in a decrease of probe pulse energy (causing lower resolution due to lower signal-to-noise ratios) with increasing pump pulse energy (lower resolution due to damage). To draw an unambiguous conclusion, this effect must be disentangled from the damage effect. Due to the scarcity of data at high pump pulse energies, the effect of increasing pump pulse energy on diffraction resolution hence remains inconclusive (Supplementary Note 4 and Supplementary Fig. 12). However, related X-ray pump/X-ray probe experiments using lysozyme microcrystals showed a clear increase of the crystal degradation with higher pump pulse energy[36].

**Effect of the X-ray pump pulse on the haemoglobin structure.** The damaging effect of a pump-pulse-induced shockwave, as indicated by reduced diffraction quality of our haemoglobin microcrystals, is consistent with a stochastic rearrangement of the crystallized molecules or unit cells affecting the order of the crystalline lattice. In addition, however, the shock wave may change the protein structure and this we also analyzed. Because the number of our diffraction images and the resolution of our data is limited (see Table 1) and because the magnitude of the structural changes are likely small, we employed a "bootstrap" resampling procedure to ensure that the observed structural differences are statistically robust (see "Methods" section and ref. [4] for a relevance of the structural differences). Haemoglobin consists of two α- and two β-subunits, each containing

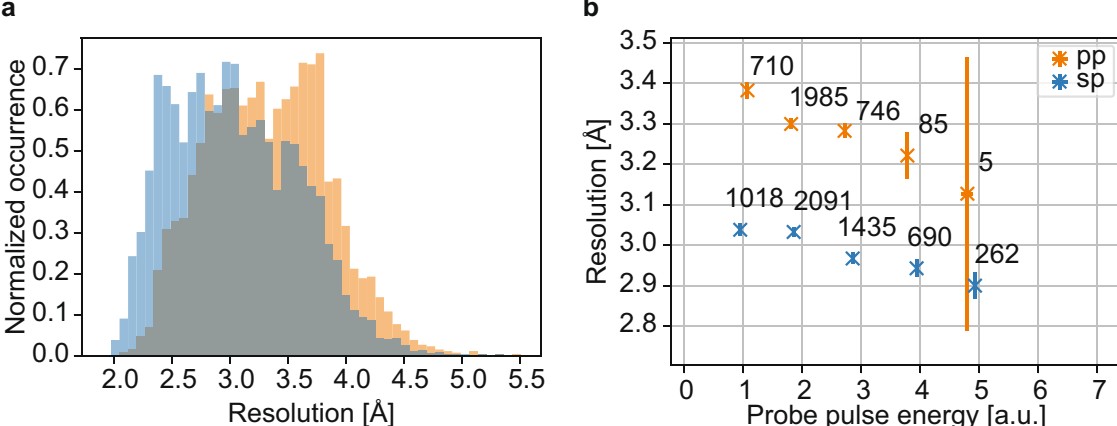

**Fig. 3 Resolution of the single-pulse and pump–probe SFX data. a** Normalized histogram of the resolution of all indexed hits of the pump–probe (orange) and single-pulse reference (blue) data set. The median resolution of all indexed hits is 3.3 Å in the pump–probe case and 3.0 Å for the single-pulse reference data. **b** Diffraction resolution of the pump–probe (orange) and the single-pulse (blue) data set as a function of probe pulse energy. Indexed haemoglobin diffraction patterns were binned according to the probe pulse energy measured by the X-ray sensitive photo diode masked with an Fe foil of same thickness as the Fe filter in front of the CSPAD detector. The median resolution of all indexed diffraction images contained in one probe pulse energy bin is plotted. The error bars correspond to the error of the mean per bin. The number next to each data point indicates the number of indexed diffraction patterns within a given bin. For all probe pulse energies, the resolution of the pump–probe data is consistently lower than the resolution of the single-pulse reference data, indicating that the observed effect is not caused by a difference in changing probe pulse energy conditions between the two data sets (Supplementary Fig. 4). Equalization of probe pulse energy distributions (Supplementary Methods) between the two data sets leads to the same conclusion (Supplementary Fig. 5). The resolution limit (**a**, **b**) corresponds to the highest resolution value of an indexed diffraction peak with a signal-to-noise ratio of $I/\sigma(I) \geq 4$.

a covalently bound haem cofactor, arranged as a dimer of α/β dimers. The four alpha-helical subunits (α1β1α2β2) enclose a large central water-filled channel that affords leeway to accommodate the large changes in the quaternary structure between the liganded R-state and unliganded T-state haemoglobin[37]. When overlaying the structures derived from the single-pulse and pump–probe data, respectively, small but significant differences in the peptide backbone become apparent (Fig. 4 and Supplementary Movies 2–4). Importantly, many of the changes are correlated along helices and connecting loops, with several amino acid residues being displaced in a similar direction. This may explain an increase in the fraction of residues involved in turns in structures derived from the pump/probe data, as opposed to single-pulse data (see Supplementary Table 1). Correlated structural displacements include, for example, a movement of the E- (α1) and F helices (β1) towards the haem; a compressive movement of the end of the E helix, the EF corner and the beginning of the F helix; of the AB corner (region between the A- and B helices) towards the E helix (β1); and of the CD corner (α1) towards the EF corner (β2) (Fig. 4 and Supplementary Movies 2–4). Analysis of the changes in pairwise distances between all Cα atoms[4,38] with respect to the single-pulse data not only implies a collective movement of the helices and the loop regions, but also a compaction of the molecule (Fig. 5 and Supplementary Fig. 13a). This is also reflected in small but significant changes in the radius of gyration (Supplementary Fig. 13b).

Collective structural changes can be both isotropic (compressive) and anisotropic (conformational)[39]. In general the compressibility of proteins is small[40,41], but a compaction of cavities has been reported for several proteins in static high-pressure experiments[42–44]. It is thus interesting to analyse whether or not shock exposure affects the central channel and cavity volumes in Hb.CO[45,46]. However, extracting channel and cavity volumes is not straightforward, in particular for large macromolecular complexes: small changes of side chain conformations can result in significant differences in cavity volumes. This is the case for some of the cavities in our Hb.CO structures, preventing

meaningful analysis (Supplementary Movie 5), in particular in view of the larger positional uncertainty of side chain compared to main chain atoms. The solvent accessible (Richards') volume of the central channel does not seem to change significantly (Supplementary Movie 5). Spectroscopic studies on carbonmonoxy myoglobin have shown a shock-induced redshift of the Soret band followed by an extended blue edge[24]. The resolution of the Hb.CO pump–probe data is not high enough to allow the detailed analysis of the haem coordination.

In conclusion, we observe small but significant differences in the structures as determined from the single-pulse and the pump–probe data sets. Confidence in these observations is heightened by the fact that the effective error bars on the observed correlated structural changes are much smaller than those for individual atoms (see also the "Methods" section on "Data processing and structure solution", as well as ref. [4]). Our conservative approach of fixing the unit cell constants likely underestimates the structural changes.

**Implications for data collection at the EuXFEL.** Our X-ray pump/X-ray probe experiments show that shockwaves generated by femtosecond X-ray pulses focused to micron-sized focal spots induce significant changes on protein crystals transported in a micron-sized liquid jet, affecting the order of the crystal lattice, as well as the protein structure. In contrast, previous studies performed at EuXFEL at 1.1 MHz repetition rate observed no such signs of shock-induced damage in SFX experiments[19,20,30,47,48]. The critically different experimental characteristics of those previous measurements are longer effective time delays (~220 ns in our case vs. ~910 ns at 1.1 MHz (refs. [19,20,30])); higher photon and pulse energies and a higher jet speed (up to 100 m s$^{-1}$) in some instances[20,30]. As an overall result of these differences, the sample probed in the 1.1 MHz measurements was subjected to significantly lower pressure pulses. The lack of shock damage observation for the 1.1 MHz experiments[19,20,30,47,48] can be explained by the rapid attenuation of the shock as it travels along the jet. Compared to our experiment, much lower shock-induced

**Table 1 Data and refinement statistics calculated for probe-only and pump–probe data.**

| Analysis approach | Same resolution limits imposed | | | Automatically determined high-resolution limits | | |
|---|---|---|---|---|---|---|
| Data set | Single-pulse | pump–probe | Single-pulse (all) | Single-pulse | pump–probe | Single-pulse (all) |
| # diffraction images used | 3500 | 3500 | 5500 | 3500 | 3500 | 5500 |
| Space group | $P2_12_12_1$ | $P2_12_12_1$ | $P2_12_12_1$ | $P2_12_12_1$ | $P2_12_12_1$ | $P2_12_12_1$ |
| Cell dimensions (Å, °) | 55.7 158.1 67.7 | 55.4 157.8 67.4 | 55.7 158.1 67.7 | 55.7 158.1 67.7 | 55.4 157.8 67.4 | 55.7 158.1 67.7 |
| | 90.0 90.0 90.0 | 90.0 90.0 90.0 | 90.0 90.0 90.0 | 90.0 90.0 90.0 | 90.0 90.0 90.0 | 90.0 90.0 90.0 |
| # indexed | 3500 | 3500 | 5500 | 3500 | 3500 | 5500 |
| Resolution (Å) | 27.85–2.50 | 27.42–2.50 | 28.48–2.50 | 28.48–2.53 | 27.42–2.77 | 28.48–2.54 |
| | (2.57–2.50) | (2.57–2.50) | (2.57–2.50) | (2.60–2.53) | (2.77–2.84) | (2.61–2.54) |
| $I/\sigma(I)$ | 2.1 (1.0) | 2.0 (0.5) | 2.4 (1.2) | 2.1 (1.0) | 2.5 (1.0) | 2.4 (1.2) |
| $R_{split}$ (%) | 48.3 (105.7) | 40.1 (235.3) | 40.0 (91.4) | 48.1 (100.3) | 36.9 (115.7) | 39.7 (89.6) |
| $CC_{1/2}$ | 0.705 (0.172) | 0.857 (0.107) | 0.8 (0.294) | 0.702 (0.233) | 0.851 (0.307) | 0.798 (0.317) |
| $CC^*$ | 0.909 (0.542) | 0.961 (0.44) | 0.943 (0.674) | 0.908 (0.615) | 0.959 (0.686) | 0.942 (0.694) |
| Completeness (%) | 99.9 (99.7) | 99.8 (99.7) | 100.0 (99.9) | 99.9 (100.0) | 99.8 (99.7) | 100.0 (100.0) |
| Multiplicity | 18.2 (12.4) | 16.4 (11.0) | 28.1 (19.1) | 18.4 (12.3) | 18.3 (11.8) | 28.6 (18.9) |
| Wilson B (Å$^2$) | 47.4 | 59.0 | 48.3 | 48.2 | 61.2 | 49.2 |
| *Refinement* | | | | | | |
| Unit cell parameters: $a\ b\ c$, $\alpha\ \beta\ \gamma$ (Å,°) | | | | 55.7 158.1 67.7, 90.0 90.0 90.0 | | |
| Images used | | | | 3500 | 3500 | 5500 |
| Resolution (Å) | | | | 28.48–2.53 | 27.42–2.77 | 28.48–2.54 |
| No. reflections | | | | 19541 | 14810 | 19339 |
| $R_{work}/R_{free}$ | | | | 0.22757/ 0.28826 | 0.18841/0.27316 | 0.20701/0.27266 |
| *No. atoms* | | | | | | |
| Protein | | | | 4332 | 4332 | 4332 |
| Ligands | | | | 172 (4 haems) 8 (4 COs) | 172 (4 haems) 8 (4 COs) | 172 (4 haems) 8 (4 COs) |
| Water | | | | 38 | 38 | 38 |
| *B-factors (Å$^2$)* | | | | | | |
| Chain A (haem A) | | | | 42.1 (35.7) | 52.4 (47.1) | 41.3 (35.5) |
| Chain B (haem B) | | | | 46.4 (41.6) | 57.8 (53.2) | 46.3 (42.4) |
| Chain C (haem C) | | | | 50.3 (40.9) | 59.9 (53.4) | 49.5 (40.4) |
| Chain D (haem D) | | | | 56.1 (45.5) | 68.5 (57.4) | 55.5 (45.2) |
| Water | | | | 41.4 | 49.6 | 42.7 |
| *R.m.s deviations* | | | | | | |
| Bond lengths (Å) | | | | 0.007 | 0.008 | 0.007 |
| Bond angles (°) | | | | 1.490 | 1.608 | 1.549 |
| *Percentage of residues in Ramachandran plot region* | | | | | | |
| Preferred | | | | 95.4 | 93.8 | 94.6 |
| Allowed | | | | 4.1 | 5.5 | 4.7 |
| Disallowed | | | | 0.5 | 0.7 | 0.7 |
| PDB code | | | | 7AET | 7AEV | 7AEU |

Values are given for either applying the same or automatically determined resolution limits, respectively, for the probe-only and pump/probe data. The first/fourth columns show the statistics of the probe data, the second/fifth for the pump/probe data using the same number of images and the third/six column the statistics of the single-pulse data, using all images available. All diffraction images that passed the filtering set were included in this analysis, irrespective of the probe pulse energy. For the automatic resolution determination, each data set was divided into 20 resolution bins, and $I/\sigma(I)$ was calculated for each bin. The resolution of the bin where this value was still >1.0 was taken as the high-resolution limit.

pressures are therefore expected for the longer distances of shockwave travel sampled in 1.1 MHz experiments[35]. This decrease is correspondingly larger when faster jets are used[20,30] (Supplementary Note 5), since the distance of shockwave travel is even larger. The pressure experienced by sample probed in our pump–probe experiments was on the order of ~40 MPa and approximately one order of magnitude higher than in the previous 1.1 MHz experiment at EuXFEL[30] (Supplementary Note 5 and ref. [36]). Moreover, and in contrast to previous published experiments[19,20,30,47], our femtosecond snapshot imaging of the jet ensures that a given sample segment has indeed experienced a shock wave. Absent this verification, small shock wave effects may be washed out by contributions from non-damaged data and so be overlooked.

Due to the two-pulse machine setup at the LCLS combined with the constrain of limiting the explosion-induced gaps such that the probe pulse hits the jet, the pump pulse energy in our X-ray pump/X-ray probe experiments was for most of the data <0.1 mJ. This pulse energy is much lower than in standard SFX experiments: a typical 4.5 MHz EuXFEL SFX experiment employs single-pulse energies of up to ~1 mJ. Since the shockwave-induced pressure jump increases with deposited XFEL energy, this would imply stronger shock-induced damage in those standard measurements. However the extent of the explosive XFEL-induced jet gap also increases with XFEL pulse energy, which requires a faster jet to avoid shooting into the gap. A faster jet increases the distance of travel of the shockwave through the jet and thus the attenuation of the shock before it encounters crystals to be probed by the next XFEL pulse. Exactly how this all balances out in 4.5 MHz operation remains to be seen. Nonetheless, we suggest that the shockwave effects may well be comparable to those reported here as discussed in detail in reference[36]. Ultimately, the effects will depend strongly on the pressure sensitivity of the sample being investigated and on the specific data collection parameters[36]. Note that jets much faster than the current 50 m s$^{-1}$

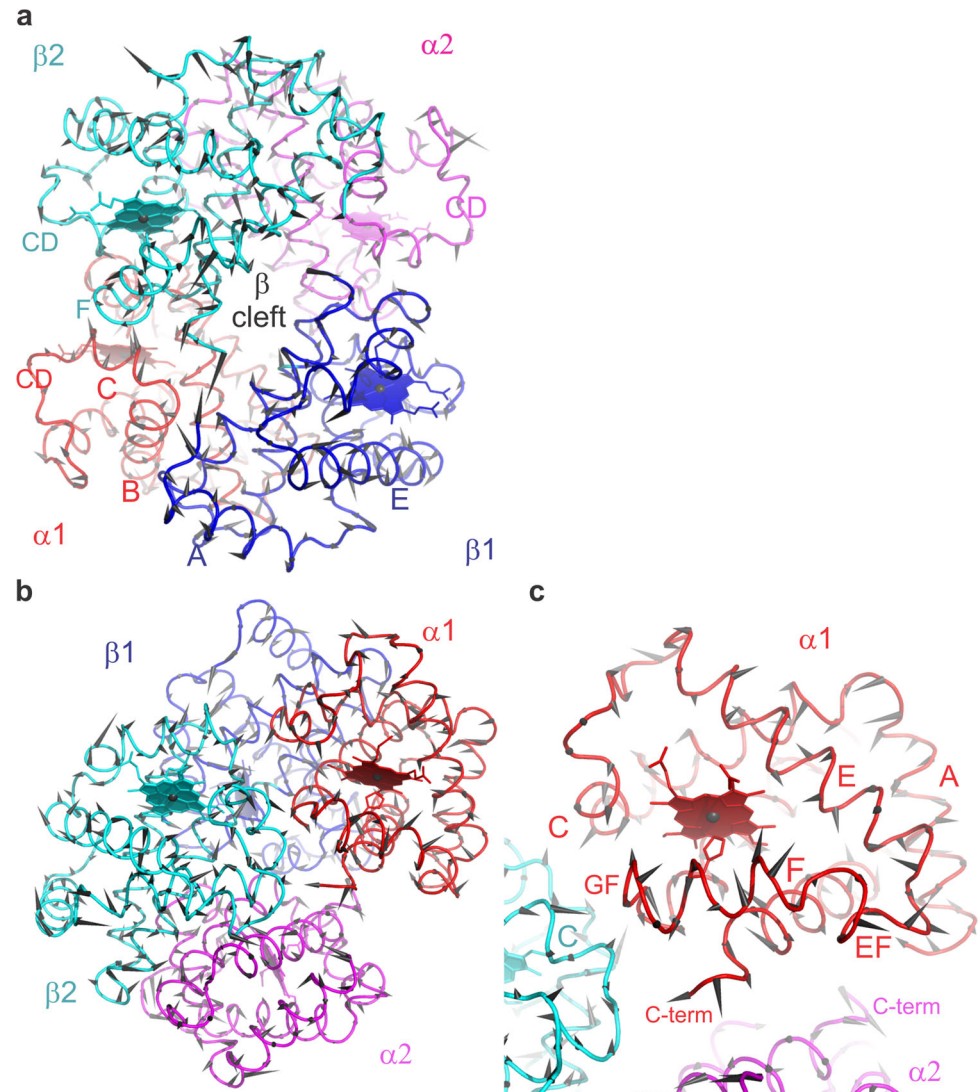

**Fig. 4 Structural comparison of the single-pulse and pump–probe Hb.CO structures.** The displacement between respective Cα positions is indicated by black arrows. The magnitude of the displacement is illustrated by the length of the arrows (multiplied by a factor of 10). **a, b** Different orientations of the haemoglobin tetramer. The fact that clusters of arrows point in similar directions shows that the displacements are correlated both within and between secondary structure elements. To ease visualization of the displacements, Supplementary Movie 4 shows a morph between the two structures. The proximal histidines are shown as sticks. **c** A magnified view of the α1 subunit (similar orientation as in **b**). Correlated displacements of the F helix towards the haem are clearly visible. **a, b, c** The alpha subunits are shown in red and magenta, beta subunits in blue and cyan. Helices are labelled by capital letters (A, B, C...), loops between helices by the two letters corresponding to the respective helices (e.g., EF loop region connecting the E and F helices); haem planes are depicted as filled planes.

will likely be required for collecting undamaged data at 4.5 MHz repetition rate.

XFEL-induced shockwaves may be relevant not just to crystalline samples, but also to XFEL scattering from solutions of proteins or other molecules[49]. While SFX experiments are typically conducted in thin microjets (diameter 3–5 μm), most spectroscopy experiments[50–52] employ large diameter Rayleigh jets (~30–100 μm) as do also many SAXS and wide-angle solution scattering experiments[50]. Again unknown balances remain to be investigated: larger jets are generally slower, likely requiring EuXFEL operation at <4.5 MHz maximum in order to clear the gap[50]. This yields an increased distance of travel and thereby a greater relative attenuation of the shock. In larger diameter jets, however, shocks also attenuate more slowly with distance. Which effect will dominate remains to be seen.

**On XFEL-induced shock effects**. As detrimental as shockwaves may be for collecting structurally valid native data, X-ray triggered shocks could also open a novel experimental regime for nanosecond time-resolved studies of, for example, pressure-induced phase transitions in liquids; pressure-induced protein unfolding; and pressure-induced pH jumps to trigger chemical reactions on rapid timescales. A sub-microsecond (0.7 μs) pressure jump instrument, described a few years ago[53], achieved pressure jumps of 0.25 GPa (2.5 kbar). The first experiments with this instrument showed refolding times of 2.1 μs in a genetically engineered lambda repressor mutant[53]. Importantly, molecular dynamics simulations supporting this result[54], as well as similar results on other systems[55], predict scientifically relevant processes on a nanosecond timescale. Since no clean, viable experimental method currently exists to explore this

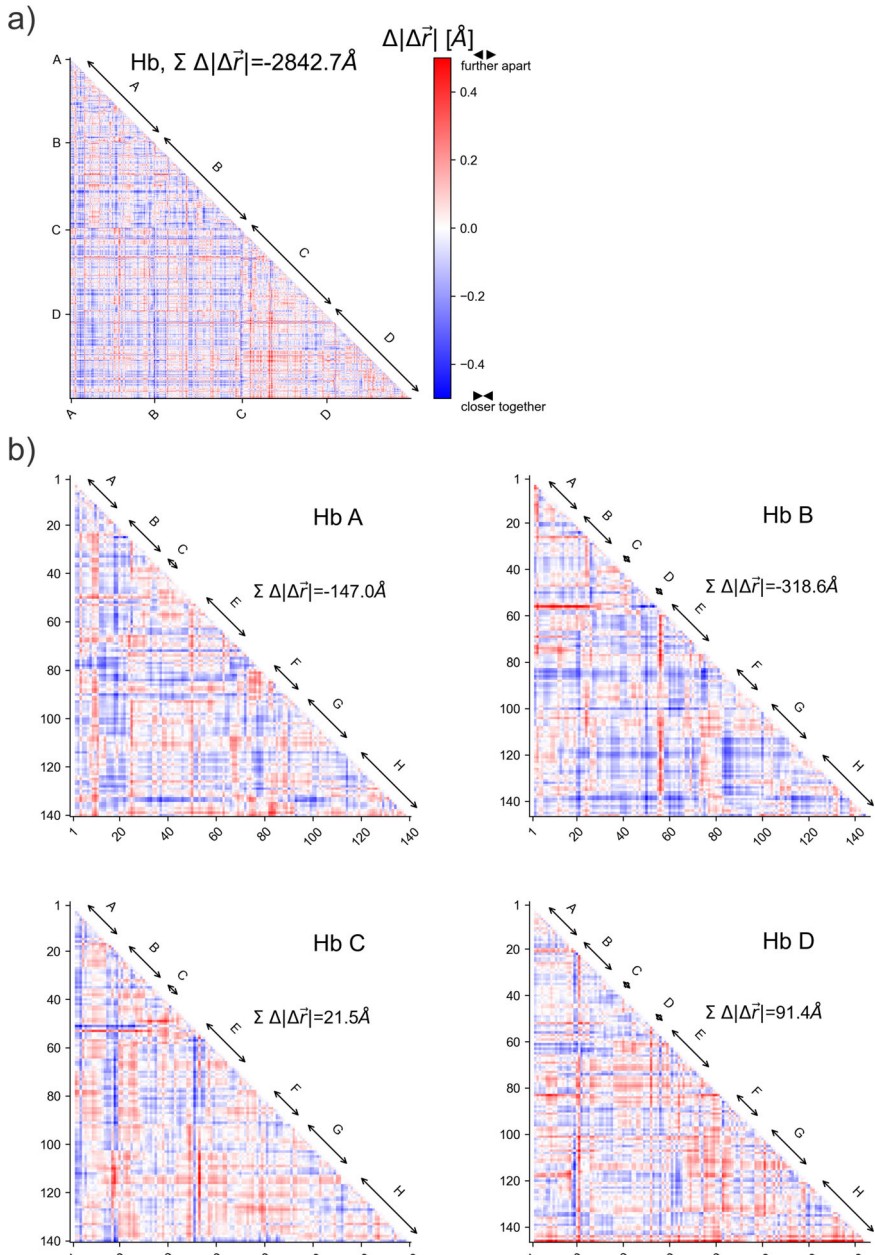

**Fig. 5 Distance matrices showing the relative displacement of Cα–Cα pairs when comparing structures determined using the single-pulse data and the pump–probe data.** Red indicates an increase of the distance, blue a decrease. The blue features dominate in the plot of the haemoglobin tetramer (**a**), as well as in the plots of the α1 (HbA) and β1 (HbB) subunits (**b**) indicating a clear compaction of the structure (see Supplementary Fig. 13). In contrast the α2 (HbC) and β2 (HbD) subunits slightly expand (**b**). Compaction and expansion are also clearly apparent from the sum of the relative changes. Correlated movements of secondary structure elements can be observed in the plots as extended streaks of blue or red colour (e.g., parts of the α1-A helix move closer to the core of the α1 subunit, similarly part of the β1 EF loop and the F helix move closer to the β1core, etc.).

important regime, the nanosecond timescale has remained largely unexplored, limiting our understanding of the fast initial steps in folding or unfolding. This experimental gap might conceivably be closed by X-ray pump/X-ray probe experiments, as described herein.

In conclusion, we here describe the first experimental evidence for shockwave-induced sample degradation and concomitant protein structural changes in MHz SFX experiments. More experiments and simulations must be performed to understand the underlying dynamics of shockwave interaction with material transported in the jet. The degree of shock-induced

crystal damage in MHz XFEL experiments (and thus the suitable XFEL repetition rate) involves a number of as yet unknown factors, including (i) the sensitivity of a particular sample to shock damage, (ii) the reversibility of the shock damage (in repetitive measurements, recovery must occur before a following measurement can be made) and (iii) the additivity of shock damage (whether crystals survive a single shock, yet deteriorate under repeated shocks). Ultimately, these properties and parameters of shock-induced damage will determine the maximum repetition rate for data collection of native samples at MHz XFELs (see also Supplementary Note 6

and ref. [36]). Although it is difficult to extrapolate the structural changes observed in haemoglobin microcrystals to other samples, shock-induced damage at 4.5 MHx XFEL operation will likely be a matter of concern in other systems.

## Methods

**Crystallization and injection.** Hb.CO was crystallized as follows[15]: human oxy haemoglobin A (Hb.O$_2$) was purified from expired units of human blood (type A) as described[56,57], and then converted to the carbonmonoxy complex. To this end, a three-neck flask was equipped with a magnetic stirring bar, two gas inlets with stop cocks and a rubber stopper, and charged with the HbO$_2$ solution. Upon repeated cycles of evacuation (5–10 min) and flushing with CO using a Schlenk line, the tomato red protein solution turned raspberry red. Neither sodium dithionite nor toluene was added. Long rod-shaped Hb.CO crystals grew in a CO saturated atmosphere at room temperature within a few days upon mixing solutions of Hb. CO (~2 mM in water) and precipitant (3.2 M NaH$_2$PO$_4$/3.2 M K$_2$HPO$_4$ in a 2:1 ratio) in a ratio of 1:2.5. Crystals were milled by filtration through a tandem array of 100–20–10 μm stainless steel filters, resulting in microcrystals of ~5 × 5 × 10 μm. The Hb.CO microcrystalline slurry (~15% (v/v) settled crystalline material) was injected by means of a GDVN at ~50 ± 5 m s$^{-1}$ producing 4–5 μm diameter jets.

**Jet imaging and speed determination.** The liquid jet was imaged from an off-axis perspective (orthogonal to both X-rays and jet flow direction) using a 50× infinity corrected objective (SL Plan Apo, Mitutoyo) in combination with a 200 mm tube lens and a camera (Opal 1000, Adimec). The magnification of the image was 0.11 μm pix$^{-1}$ and the optical resolution was measured with a resolution target (Extreme USAF resolution target, Ready Optics) to be better than 700 nm. To obtain sharp images without motion blur, frequency-doubled light pulses (400 nm) from the femtosecond laser system at CXI[32] were employed to illuminate the jet. Laser and camera were triggered to record images of the jet at chosen time delays relative to the X-ray pulses. For all data contained in this analysis, images were taken within a few nanoseconds after the second X-ray pulse, thus imaging the effect of the first two pulses on the jet.

Due to their small diameter (~5 μm), the optical path difference induced by the shock compression is smaller in our jets than in previously imaged jets (14–30 μm diameter)[22,35] and the shocked region does not deflect the illumination light sufficiently to make the shocks visible. Accordingly, the shockwaves in our jets could not directly be imaged reliably.

The centre of the XFEL-induced gap in the jet moves downstream with the speed of the jet. To track the movement of gaps over time, the centre of the gap $y$ was measured at two different imaging time delays $t_1$ and $t_2$. Jet speed is then obtained as the distance between the gap centres at both time delays, divided by the temporal separation of the two imaging delays: $v = (y(t_1) - y(t_2))/(t_1 - t_2)$.

**Data collection.** The experiment was performed at the CXI instrument at the LCLS. Two pulses separated by 122.5 ns interrogated the liquid jet. The two pulses were focused to two points vertically separated by 5 μm, with the first (pump) pulse intersecting the liquid jet further downstream than the second (probe) pulse (Fig. 1). The beam size was ~1.5 × 1.5 μm² (FWHM) for both pulses. The photon energies of the pump and probe X-ray pulse (~0.9 mJ combined pulse energy, pulse duration ~30 fs) were set to ~40 eV above and below the iron K-absorption edge (~7.11 keV), respectively. A 25 μm thick iron foil in front of the CSPAD detector absorbed the pump but not the probe pulse[33,34] (Fig. 1) such that only diffraction patterns generated by the probe pulse were recorded.

The two-bunch mode exhibits large fluctuations in the pulse energies, and is also more susceptible to photon energy drifts than the commonly used SASE single-pulse mode[58]. The LCLS gas detector cannot resolve pulse energies with nanosecond time resolution. To ensure that the two pulses have the expected photon energies and to enable analysis of the data as a function of pump and probe pulse energies, we used diagnostics based on two fast photodiodes (Hamamatsu MSM, 30 ps response time) picking up X-rays scattered by a Kapton foil. The diode signal was recorded synchronously with the CSPAD data. One diode was covered with a 25 μm Fe foil like the detector. The non-masked diode measured the relative pulse energy of each pulse, evaluated as the integrated signal of each pulse. Comparison with the signal of the masked diode allowed checking whether the pump pulse photon energy was indeed above the Fe K-edge, thus not erroneously contributing to the measured diffraction signal due to the probe pulse.

**Data filtering conditions.** Prior to further analysis, diffraction data was filtered with respect to the two diode signals to exclude data in which diode signals were affected by electronic noise. To exclude noisy shots, the standard deviation of the voltage trace was calculated over a time interval preceding the arrival of pulses, and shots were excluded if this was larger than the mean of the standard deviations of noise-less traces plus three times the corresponding standard deviation. For pump–probe data, the diode signals were also analysed to exclude hits where the pump pulse signal on the masked diode was high, potentially indicating (partial) leakage of the pump photon energy below the Fe K-edge. To find this threshold, the full trace of the diode signal was plotted for different pump pulse signals. Hits in

which the pump pulse signal was smaller than the mean pump pulse signal of all shots plus ~1.5 times its standard deviation did not exhibit any visible pump pulse signal and were used for analysis.

Images of the jet, recorded a few nanoseconds after impact of the probe pulse onto the jet, were analyzed for each pump–probe shot to determine if the pump pulse launched a shock wave affecting sample interrogated by the X-ray probe pulse. For this purpose, a custom-written python script analyzed jet shape, determining location and size of gaps in the jet, as well as the size of the jet projected onto the horizontal plane. The latter indicates whether the angle of the jet to the X-ray beam axis has changed or whether the jet had a different diameter (which impacts the magnitude and the decay of shockwaves). The location and size of gaps in the jet help reveal if the pump pulse interacted with the jet, and whether it launched a shock wave that could propagate to the jet region probed by the second X-ray pulse. Even if the pump pulse hit the jet, the desired propagation of the shock wave through the jet is precluded if the pump pulse strikes downstream of the jet break-up point. This situation can be identified based on the location of gaps in the jet. For shock analysis, only those indexed hits were used that do not show any abnormal jet morphology (projected jet size within one standard deviation of the median projected size within a given run), in which one or two gaps due to pump or pump and probe pulse are clearly visible (a gap due to the probe pulse does not necessarily need to exist as long as the probe pulse leads to an indexable diffraction pattern recorded by the detector) and in which the jet is fully continuous upstream of the pumped segment (thus allowing propagation of the shock wave upstream). A more detailed protocol of this filtering procedure is described in the Supplementary Methods. The same filtering effect (ensuring that a shock wave had been launched by the pump pulse) as obtained by filtering on the jet images can be achieved by filtering based on the detector images, if these are saved by a fast detector for all pump and probe pulses like for example at EuXFEL[48].

**Data processing and structure solution.** Online data analysis was performed with CASS[59]. A diffraction pattern was considered a hit if it contained ≥10 peaks. To evaluate the resolution of single diffraction images at a given signal-to-noise threshold $x$, custom-written python scripts evaluated the best resolution of all indexed reflections with $I/\sigma(I) \geq x$. The detector metrology was optimized in two steps: after optimization of the detector panel alignment (see ref. [4]) the distance between the detector and the XFEL interaction zone was optimized by a parameter grid search minimizing the root mean square deviation between reflections and diffraction peaks, as measured by the geoptimiser tool from the CrystFEL software suite, version 0.8.0 (refs. [60,61]). Diffraction peaks were identified by CrystFEL from calibrated detector images that had passed filtering using the gradient search after Zaefferer, with the following peak detection parameters: --threshold=70 --min-snr=5 --min-gradient=10000 --tolerance=10,10,10,2 --median-filter=16. Indexing and integration of the filtered diffraction images were performed with CrystFEL using the xgandalf, dirax and mosflm indexers with the no-cell-combinations option. The indexed data set was merged using process_hkl from the CrystFEL software suite without scaling or partiality correction. The Hb.CO pump-only and pump–probe data were phased by molecular replacement with PHASER[62] using PDB entry 6HAL as the search model[15], and refined using alternating cycles of rebuilding in COOT[63,64] and refinement in REFMAC5 (ref. [65]). For both probe-only and pump–probe structures a final round of refinement using identical parameters was performed, to ensure comparability. Moreover, the same (probe-only) unit cell parameters were imposed during refinement of all structures because of uncertainties in the probe photon energy during the pump–probe experiment. Data and model statistics are given in Table 1, the quality of the computed electron density map is shown in Supplementary Fig. 14. The structures were used as starting models for the refinement of the data using a resampling method to estimate the coordinate uncertainties[34]. In this case, we used bootstrapping, and prepared 100 resampled data sets by randomly drawing images with replacement from the pool of available images for a data set until the same number of images was reached as was available for the original data set. Against these resampled data sets, 100 structures were then refined using REFMAC, starting with rigid body refinement of all four monomers, and finally restrained refinement of all atomic positions and B-factors. The standard deviation of the ensemble-averaged boot-strapped structures gives the mean error of the coordinates; the structures determined using the original sets of diffraction images (i.e., without resampling) are within two sigma from the corresponding averaged positions, i.e., within the 90% confidence level. These structures were used for the detailed analysis, with the bootstrapped ensembles yielding the positional errors of the respective coordinates. The structural differences between the pump-only and pump–probe data were visualized using the modevector.py script from Pymol[66], as described previously[43,67]. Channel and cavity volumes were analysed with CastP3.0 (ref. [68]) and 3V[46]. The magnitude of the observed structural changes between the single-pulse and pump–probe structures (both those determined using all images and the ensemble-averaged ones), respectively, is small (r.m.s.d. 0.11 Å for 552 aligned residues). This is similar to the expected coordinate error given by the average displacement/atom in the ensemble-averaged structures (0.12 Å over 4550 atoms). However, many of the changes are correlated along secondary structure elements. This decreases the effective error bars on the observed correlated changes, making them much smaller than those for individual atoms (see also ref. [4]). It is unlikely

that the effects observed in our experiments are caused by radiation damage: due to the vertical displacement of the pump and the probe interaction region on the jet, jet segments probed by the second pulse were not illuminated directly by the first pulse (see Supplementary Note 3).

**Reporting summary**. Further information on research design is available in the Nature Research Reporting Summary linked to this article.

## Data availability

Coordinates have been deposited with the PDB 7AET, 7AEU, 7AEV, the indexed images are deposited at CXIDB.org[69], ID 143; https://doi.org/10.11577/1616145 . Other data are available from the authors upon reasonable request.

## Code availability

Analysis scripts are available from the authors upon request.

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

## Acknowledgements

The experiments were performed at the Linac Coherent Light Source (LCLS), SLAC National Accelerator Laboratory. Use of the LCLS is supported by the U.S. Department of Energy, Office of Science, Office of Basic Energy Sciences under Contract no. DE-AC02-76SF00515. The research was supported by the Max Planck Society and startup funds from Rutgers University Newark to C.A.S. Part of the sample injector used at LCLS for this research was funded by the National Institutes of Health, P41GM103393, formerly P41RR001209. We thank Yaroslav Aulin and Piotr Piotrowiak for assistance in testing the diode system.

## Author contributions

C.A.S. and I.S. conceived the experiment, which was designed and coordinated by C.A.S., F.J.D, I.S., L.F., M.L.G. and S.B.; E.H., R.L.S. and I.S. prepared samples; W.C., S.G., J.L., A.L., J.S., S.V. and F.J.D setup and executed the two-pulse FEL mode; C.A.S. setup femtosecond imaging and (with L.F.) diode-base pulse diagnostics; M.L.G., R.B.D., M.K., M.S., G.N.K. and R.L.S performed sample injection, A.A., S.C., M.H., J.E.K., T.J.L., T.V.D. and S.B. operated CXI and collected data; M.H., K.N., C.M.R. and L.F. performed online processing; L.F. and A.G. performed offline data processing; A.G., M.L.G., C.A.S., I.S. and T.R.M.B. analysed diffraction data. T.R.M.B supervised the crystallographic analysis. All authors jointly performed the experiment, discussed the results and contributed to the manuscript. The initial versions were written by M.L.G., C.A.S., R.B.D. and I.S.

## Funding

## Competing interests

The authors declare no competing interests.
