## [Peer Review File · Nature Communications]

REVIEWER COMMENTS

Reviewer #1 (Remarks to the Author):

This paper raises the issue of new sample damage in SFX experiments caused by shock waves generated by MHz XFEL pulses at microsecond intervals and describes the preliminary verification results of the damage using hemoglobin microcrystals.

Macromolecular crystallography has made remarkable progress in recent years, such as enabling structural analysis from microcrystals of membrane proteins by high-brilliant synchrotron radiation and become an important tool for life science research. However, the use of high-brilliant synchrotron radiation causes the serious radiation damage for protein crystals which make detailed structural and functional studies of protein functions difficult. Since 2010, XFEL, which can utilize high-intensity short X-ray pulses in femtoseconds, has opened a new field for radiation-damage-free structural analysis and dynamic structural analysis using the "diffraction before destruction" manner. SFX, which continuously supplies samples to XFEL pulses with a liquid jet, is widely used in XFEL's macromolecular crystallography. In the low-repetition XFEL pulses of non-superconducting XFEL, the sample amount used in the experiment is enormous compared to the sample amount that actually contributes to the diffraction measurement. Reducing the sample consumption and improving the efficiency of data collections are big issues for current SFX. Therefore, superconducting XFEL has been developed and MHz XFEL pulses at sub-microsecond intervals will become available. In SFX, it was expected that almost all samples would contribute to the diffraction measurement efficiently, but there was a new concern about sample damage caused by the liquid explosions induced by X-ray laser pulses.

The authors conducted an SFX experiment using simulated 4.5MHz XFEL pulses in the "two bunch" mode of the existing XFEL (LCLS) and observed the effect of shock wave caused by the liquid explosions on a crystal sample. From the experiment, the authors clarified that XFEL pulses with 4.5 MHz repetition caused deterioration of crystal quality and structure change, which was not seen with 1.197 MHz repetition. The authors conclude that the shock waves generated by XFEL reduce the crystal hit rate, worsen the crystal quality evaluation indices, and reduce the maximum resolution. These conclusions may require further investigation, but are very interesting as early experimental results.

It is also noteworthy that the author proposes X-ray triggered shocks could also open a novel experimental regime for nanosecond time-resolved studies by taking advantage of the influence of the pressure to the protein structure due to the shock wave.

Although questions remain on some of the detail points mentioned below, I recommend this paper should be published early as the result of the first SFX experiment assuming EuXFEL repetition at 4.5 MHz, and as an important finding that further condition studies are needed for shock wave sample damage in future MHz XFEL pulse experiments.

Questions and comments about individual items

1. In this paper, authors cite the 2015 Stan's Nature Physics paper in order to describe that shock waves are generated by Liquid explosions induced by XFEL pulses. Is this phenomenon the same as the ionization of molecules by high-intensity X-ray pulses and the resulting Coulomb explosion in Neutze's Nature paper in 2000? Since it is described that Coulomb explosion occurs in XFEL in the Neutze's Nature paper, it would be helpful for the reader to have a brief explanation of the relationship between liquid explosion and Coulomb explosion.

2. Figure 1. Experimental setup.

In addition to the experimental setup, Fig. 1 should be more clearly understandable by enlarging the shock wave part. It may also be effective to add a photomicrograph of the shock wave generated by XFEL pulse from Fig. 2 of the supplementary material.

3. In comparison the diffraction data using the single pulse and pump-probe data, are the average hit rates of 19% and 13% for SP and PP the values before the filtering analysis, respectively? The author concludes that the decrease in hit rate is due to the deterioration of the crystal due to the shock wave, as there is no change in the stability of the crystal or jet. However, the number of images before and after the filter analysis in the pump-probe data dropped significantly, and it

would be suspected that the hit rate could decrease due to the gap and/or the disturbance of the stream induced by the liquid explosion or the shock wave.

Please consider a more detailed examination and description of the state of the jet stream in the pump-probe experiment. If jet stream disruption at the XFEL pulse point is suspected, the potentials of jet stream disruption as well as crystal quality decrease against reduced hit rates should be described.

Although there is no big difference in the index rate in the filtered data, it is very meaningful information that the deterioration of crystal quality can be confirmed in the data processing result. For more accurate understanding, the total number of X-ray shots and indexed images at the data collection should be described in the footnote of Fig. 2 or the text.

4. Figure 2. Hit and indexing rates in pump-probe and probe-only runs.

a). The hit rate pump-probe has data points, but b) the Indexing rate has only 6 data points plotted. Why are there two fewer data points in the hit rate graph?

5. In the radiation damage of synchrotron radiation, the local damage of protein molecules due to the chemical reaction by high-energy electrons and radicals generated by X-ray becomes a problem. In this sub-microsecond pump and probe experiment, the time scale is sufficient to that the above chemical reaction proceeds. From the flight distance of each particle, it seems that the two XFEL pulse irradiation points are sufficiently separated, but did you see any local damage to the side chains of the refined structure of the hemoglobin? In order to clarify the distinction between shock wave damage, which is the subject of this paper, and radiation damage that has been known so far, I think it is better to briefly describe the radiation damage of the results of this structural analysis.

Reviewer #2 (Remarks to the Author):

In this work, Grünbein et al. evaluated the influence of so-called shockwave during serial femtosecond crystallography (SFX) measurements on protein crystal samples using the properties of LCLS with a view to developing a measurement method in EuXFEL.

Although disturbance of the sample stream from the injector by shockwave has conventionally been considered only an obstacle in SFX measurements, the authors precisely analyzed the properties of shockwave.

In doing so, the authors not only contributed to the establishment of new techniques for SFX measurement methods, but also presented fundamental findings for the future development of novel analytical methods for understanding biochemistry, such as protein folding.

I would like to request that the authors answer the following questions/comments.

About Figure 1, the position of the GDVN nozzle should be clearly indicated so that the direction of the sample flow can be recognized at a glance, because in the current Fig, non-specialists may think that the sample flows from bottom to top.

How about also describing the length between the sample flow and the iron filter?

The overall setup of the instrumentation described in the Methods section appeared to be elaborate. However, since the layout of the instruments is difficult to understand in a text description alone, why don't authors provide supplementary photos of the actual instrumentation so that it can be easily reproduced/verified by other researchers?

What is the focused beam size (400 nm?) of the XFEL pump/probe? It should be clearly stated in the Methods section.

Why is the resolution of haemoglobin structure so poor in this study, even in single-pulse scheme data? Simply because of low pulse energy? I had imagined that haemoglobin crystals diffract to high resolutions more than 2 Å.

To allow the reader to visually evaluate the quality of the solved structure, the authors must include electron density maps as supplementary Figures. In particular, I would like to see the electron density around the haem cofactor.

I am concerned that the crystals/proteins could have been damaged not by shockwave, but by pump XFEL's scattered light that reached the probe position. Is this possibility ruled out? (Even if it's due to scattered light, I think it's a worthwhile study.)

I think the degree of conformational doming of haem (a curvature of the haem plane) varies with photolysis or metal redox state within 1 ns period (e.g., Levantino et al. Struct. Dyn. 2015). By comparing the conformations of haem cofactors between single-pulse data and pump-probe data, it may be possible to see if photolysis/photoreduction has occurred (although the resolution for that may be insufficient). Alternatively, the authors may be able to test whether photoreduction occurred using other photosensitive samples instead of hemoglobin crystals in the future work.

Minor points:

Page 4, line 90

"Such "packing defects" and the large solvent channels in protein crystals render proteins and their crystals sensitive to applied pressure."

Please cite references if available.

Page 12, line 346

"such that only diffraction patterns generated by the pump pulse were recorded."

the pump pulse -> the probe pulse

Page 14, line 395

"the distance between thee detector and the XFEL"

thee -> the

In Table 1, Rwork/Rfree for pump probe in Automatically determined high resolution limits
0.18841 (0.27316) -> 0.18841 / 0.27316

Eiichi MIZOHATA

Reviewer #3 (Remarks to the Author):

Under a specific set of conditions, the experiments reported by Grünbein et al are intended to assess the impact of shock waves traveling up a liquid column that were created by an XFEL X-ray pulse. In these results, the shock wave is assumed but not visualized as limited by the small diameter of the liquid jet. The timing delay and offset between the pump and probe are 122.5 ns and ~ 5 µm upstream in the liquid jet, respectively. These conditions probe an analogous region of the jet anticipated by the 4.5 MHz Eu.XFEL conditions, with the caveat that the delay between the pump and probe is shorter by about half. The shock wave(s) travels at supersonic speed and will have passed by this region of the jet before the region is probed with the second X-ray pulse.

This is the first of this type of experiment. These are very interesting results and should be published, after considering the comments below.

The temporal response to the pressure jump is probed with only one time point (~ 122 ns) and with only one type of sample (haemoglobin CO complex, Hb.CO) which does demonstrate changes in the atomic models fit to the pump-probe SFX diffraction data. The authors should include a statement on the estimated errors of the atomic models as these are likely to be similar to the magnitude of the changes observed in these pump-probe results. In contrast, and as briefly stated by the authors (lines 242 - 250), experiments at the Eu.XFEL reported by Yefanov et al ((2019) Struct Dyn 6, 064702) indicate that with a 100 m/s liquid jet and a 1.1 MHz pulse frequency, there is no observable difference in atomic models of hen egg white lysozyme refined against SFX datasets derived from each of the sequential pulses within the 1.1 MHz pulse train. The differences between experimental conditions between the two reports are rather significant (please see the attached image -- Illustrated to approximate scale) and should be discussed further. Indeed, Grünbein et al state that their use of a 50 m/s jet speed is the "minimum allowable" for SFX measurements collected with MHz pulse frequencies at the European XFEL. In fact, their data presented herein suggests that SFX data collection at MHz XFEL sources should probably be carried out with jets faster than 50 m/s.

The pump-probe SFX diffraction data was obtained from rod-shaped Hb.CO crystals that were of similar size to the jet diameter and the offset of the two X-ray pulses. This reviewer looked for, but could not find, the size of either the pump or probe X-ray beams; both of these must be included. The crystal density in the slurry was not listed and should be given. There was a statement (supplementary note 2; "flow alignment of the Hb.CO microcrystals in the liquid jet") that there may be a preponderance of the long axis of the rods aligned with the jet stream. Thus, it is also possible that a 10 μm long crystal rod could be hit by the weaker pump pulse and then again by the stronger probe pulse. If so, then these crystals might exhibit reduced diffraction quality due to more traditional radiation damage mechanisms, rather than the presumed pressure wave(s). It is unlikely that the observed and reported overall statistical differences will derive from a relatively small number of examples that may fall into this particular scenario. Without more information on the crystal density and estimate of the propensity to flow align, one cannot estimate the potential impact to the experiment.

The use of the two-bunch mode at the LCLS does impact the experiment since the pump pulse had significantly lower photon pulse energy and impacting the resulting pressure waves compared to the probe pulse. The stated ~ 0.03 mJ (line 470) on average and up to ~ 0.1 mJ per pump pulse is significantly lower than the ~ 2 mJ per pulse in earlier studies (especially Stan et al (2016) Nat Phys 12, 966-971) that characterized the shock waves in larger diameter jets. However, the authors present femtosecond snapshot imaging results that indicate that the pump pulse does cut the jet and by extension also likely induces shock waves along the liquid column. It is not clear how these different experimental conditions translate into pressure waves impacting crystals within the slurry of the liquid jet.

Lines 158-159 and Table 1

The authors state, "We collected 43,003 and 25,742 hits, respectively, for the pump-probe and single-pulse setups." Although, "hits" should probably be more explicitly defined as crystal lattices, the implication is that they have rather robust and complete datasets. A smaller number of these passed the various filter analysis tests (14,434 and 24,083). Thus, the pump-probe experimental regime resulted in retention of only $\sim 33\%$ of the lattices compared to $\sim 94\%$ of the single-pulse experiments. Then in Table 1, there are only 3500 "diffraction images used" which again should probably be crystal lattices and is only about 8% of original pump-probe "hits". The data only extends to modest resolution (2.54 – 2.77 \AA) even with liberal cut-off of parameters. Although the authors discuss a desire to compare datasets from as similar conditions as possible, it is not at all clear where in the process such a large fraction of the lattices get rejected?

Line 118 and line 717 (Ref 28).

Reference 28 is very a hard to find; please add another more accessible reference to assist the readers.

Line 341:

Please indicate the approximate size of the X-ray beams at both the pump and probe locations.

Line 542 " β 1core, ...)."

The use of three dots makes it appear that this legend has not been completed?

Response to Referees

We thank the three referees for their careful reading of the manuscript and helpful comments that have improved the revised manuscript. We address the specific comments of each referee below. Since all referees addressed radiation damage as a potential cause for the observed decrease of diffraction quality of data collected with the second (probe) pulse, we here summarize our reasons, arguing against this possibility. We have also addressed the potential issue of radiation damage in the main text and the supplement (Supplementary Note 3).

There are two scenarios how the first pulse could inflict radiation damage on sample probed with the second pulse:

- 1) Stray light from first pulse
- 2) Photoelectrons, radicals created by first pulse

Given the setup of the experiment, the probability is extremely low for both scenarios.

The probability of the pump beam hitting the downstream end of the crystal (so that the probe beam can intersect with the upstream end of crystal) depends on the length of the crystals along the jet axis, the distance between pump and probe interaction region, and the distance traveled by the jet between pump and probe pulse.

The pump and probe X-ray beams ($\text{\AA} \sim 1.5 \mu\text{m}$ (FWHM) each) are separated by $5 \mu\text{m}$, moreover, the jet (including the crystals) moves $\sim 6 \mu\text{m}$ during the 122.5 ns time delay between the two pulses (jet speed 50 m/s). Thus, the distance between the first pump XFEL shot and the second probe XFEL shot along the jet is $\sim 11 \mu\text{m}$, which is slightly larger than the longest crystal dimension. Thus, a crystal that is hit at its front (downstream) end by the pump pulse is located $1 \mu\text{m}$ below the probe pulse interaction region at the time of the probe pulse arrival. Therefore, radiation damage can only be induced either directly by the “wings” of the focused pump beam or indirectly by photoelectrons or diffusing radicals. The latter are too slow to reach the upstream end of the crystal before the probe beam hits, 122.5 ns after the first pulse. The free path length of photoelectrons is small and strongly depends on their energy and thus the photon energy ($2 \mu\text{m}$ for 13.5 keV , $4 \mu\text{m}$ for 18.5 keV see Nave & Hill, *J Syn. Rad.* 12, 299-303, (2005); Sanishvili et al, *PNAS* 108, 6127-6132 (2011); de la Mora et al., *PNAS* 117, 4142-4151 (2020)) and is negligible in our experiment for $\sim 7 \text{ keV}$ photons. Thus, they too are highly unlikely to cause damage. Assuming a Gaussian beam profile, the intensity of the XFEL wings at the probe position ($11 \mu\text{m}$ upstream of the pump pulse) is $\sim 10^{-52}$ of that of the pump beam at the focal position and thus negligible. Therefore, also the “stray” X-ray photons are extremely unlikely to induce any measurable damage.

Reviewer #1 (Remarks to the Author):

This paper raises the issue of new sample damage in SFX experiments caused by shock waves generated by MHz XFEL pulses at microsecond intervals and describes the preliminary verification results of the damage using hemoglobin microcrystals.

Macromolecular crystallography has made remarkable progress in recent years, such as enabling structural analysis from microcrystals of membrane proteins by high-brilliant synchrotron radiation and become an important tool for life science research. However, the use of high-brilliant synchrotron radiation causes the serious radiation damage for protein crystals which make detailed structural and functional studies of protein functions difficult. Since 2010, XFEL, which can utilize high-intensity short X-ray pulses in femtoseconds, has opened a new field for radiation-damage-free structural analysis and dynamic structural analysis using the "diffraction before destruction" manner. SFX, which continuously supplies samples to XFEL pulses with a liquid jet, is widely used in XFEL's macromolecular crystallography. In the low-repetition XFEL pulses of non-superconducting XFEL, the sample amount used in the experiment is enormous compared to the sample amount that actually contributes to the diffraction measurement. Reducing the sample consumption and improving the efficiency of data collections are big issues for current SFX.

Therefore, superconducting XFEL has been developed and MHz XFEL pulses at sub-microsecond intervals will become available. In SFX, it was expected that almost all samples would contribute to the diffraction measurement efficiently, but there was a new concern about sample damage caused by the liquid explosions induced by X-ray laser pulses.

The authors conducted an SFX experiment using simulated 4.5MHz XFEL pulses in the "two bunch" mode of the existing XFEL (LCLS) and observed the effect of shock wave caused by the liquid explosions on a crystal sample. From the experiment, the authors clarified that XFEL pulses with 4.5 MHz repetition caused deterioration of crystal quality and structure change, which was not seen with 1.197 MHz repetition. The authors conclude that the shock waves generated by XFEL reduce the crystal hit rate, worsen the crystal quality evaluation indices, and reduce the maximum resolution. These conclusions may require further investigation, but are very interesting as early experimental results.

It is also noteworthy that the author proposes X-ray triggered shocks could also open a novel experimental regime for nanosecond time-resolved studies by taking advantage of the influence of the pressure to the protein structure due to the shock wave.

Although questions remain on some of the detail points mentioned below, I recommend this paper should be published early as the result of the first SFX experiment assuming EuXFEL repetition at 4.5 MHz, and as an important finding that further condition studies are needed for shock wave sample damage in future MHz XFEL pulse experiments.

Questions and comments about individual items

1. In this paper, authors cite the 2015 Stan's Nature Physics paper in order to describe that shock waves are generated by Liquid explosions induced by XFEL pulses. Is this phenomenon the same as the ionization of molecules by high-intensity X-ray pulses and the resulting Coulomb explosion in Neutze's Nature paper in 2000? Since it is described that Coulomb explosion occurs

in XFEL in the Neutze's Nature paper, it would be helpful for the reader to have a brief explanation of the relationship between liquid explosion and Coulomb explosion.

The referee is right that Coulomb explosions and liquid jet explosions are somehow related, but they are not the same. Material exposed to intense XFEL pulses is very rapidly highly ionized, resulting in formation of a hot plasma. The positively charged ions repel each other (photo-electrons have escaped) and very small samples, such as "naked" single particles explode very rapidly, a process dubbed Coulomb explosion. Larger samples, such as particles surrounded by a water shell, supply a bath of photo-induced free electrons to the sample and slow the hydrodynamic expansion through initial confinement, acting as a tamper. Nevertheless, the sample is hot and charged and will eventually disintegrate.

Concerning our experiment, first, the target is larger than the X-ray beam, so the energy deposited is temporarily confined by the rest of the target, and second, this energy is transmitted partially into the rest of the target, and a significant fraction of this energy propagates as shock waves in the target. The observed jet explosion is rooted in isochoric heating of the water jet. In conclusion, in liquid jets one does not observe Coulomb explosions. We consider the distinction of the two processes beyond the scope of this manuscript.

2. Figure 1. Experimental setup.

In addition to the experimental setup, Fig. 1 should be more clearly understandable by enlarging the shock wave part.

We modified Fig. 1 accordingly.

It may also be effective to add a photomicrograph of the shock wave generated by XFEL pulse from Fig. 2 of the supplementary material.

We are confused to which Fig. 2 the referee refers to. As mentioned (ll. 328-331), for lack of contrast, the shock waves cannot be imaged in the small microjets used for SFX. Shocks would be visible only for stronger shock or thicker jets. The supplementary movies in Stan et al (2016) show that shocks were barely observable in $\sim 5.5 \mu\text{m}$ diameter water jets and 1.5 mJ pulse energy at source – which is at least a 10 times higher pulse energy as available for the experiment described in this manuscript.

We agree with the referee though that Figure 1 could depict the shock waves more explicitly. We adapted the schematic drawing of the setup along those lines and included also a photomicrograph of the jet, showing 2 sites of explosion due to interaction with the pump and probe pulse.

3. In comparison the diffraction data using the single pulse and pump-probe data, are the average hit rates of 19% and 13% for SP and PP the values before the filtering analysis, respectively?

We thank the referee for catching the omission. Indeed, the average hit rates are evaluated on all data before the filtering analysis, because only the jet images of crystal hits (not of all X-ray shots) have been analyzed for the pump-probe data set. We have now clarified this in the revised version of the manuscript, both in the main text and in Figure 2, where we changed the caption for panel a from “all hits were taken into account” to “all hits prior to filtering were taken into account”.

The author concludes that the decrease in hit rate is due to the deterioration of the crystal due to the shock wave, as there is no change in the stability of the crystal or jet. However, the number of images before and after the filter analysis in the pump-probe data dropped significantly, and it would be suspected that the hit rate could decrease due to the gap and/or the disturbance of the stream induced by the liquid explosion or the shock wave. Please consider a more detailed examination and description of the state of the jet stream in the pump-probe experiment. If jet stream disruption at the XFEL pulse point is suspected, the potentials of jet stream disruption as well as crystal quality decrease against reduced hit rates should be described.

The experiment was set up to observe two distinct gaps, which generally rules out making gaps that are too large. But the explosions of badly-shaped jets are more complicated and cannot be predicted based on what we know about XFEL explosions. Therefore, an impact on the hit rate is possible. Hence, the referee is correct in that we cannot exclude that the decrease in hit rate at higher pump pulse energies is solely due to the decrease in crystal quality, an additional factor may be the increasing jet disruption. We clarified this in the text.

Although there is no big difference in the index rate in the filtered data, it is very meaningful information that the deterioration of crystal quality can be confirmed in the data processing result. For more accurate understanding, the total number of X-ray shots and indexed images at the data collection should be described in the footnote of Fig. 2 or the text.

We agree with the referee that this is important information. The number of hits before and after filtering is listed in the manuscript (beginning of section “Effect of the X-ray pump pulse launched...”). We now also added the total number of indexed images of each data set after filtering.

4. Figure 2. Hit and indexing rates in pump-probe and probe-only runs.

a). The hit rate pump-probe has data points, but b) the Indexing rate has only 6 data points plotted. Why are there two fewer data points in the hit rate graph?

We thank the referee for catching this. For two runs (run 80 and 85) the jet images could not be analyzed because the time delay of the jet imaging varied during those runs. Therefore, these images could not be used to decide whether or not a shock wave had been launched that successfully propagated to the jet segment to be probed. We therefore excluded all data from these runs from our analysis. We clarified this in the caption of Figure 2.

5. In the radiation damage of synchrotron radiation, the local damage of protein molecules due to the chemical reaction by high-energy electrons and radicals generated by X-ray becomes a problem. In this sub-microsecond pump and probe experiment, the time scale is sufficient to that the above chemical reaction proceeds.

We are not aware of publications describing ns radical chemistry particularly in (crystalline) proteins. But we agree, in principle it might happen. However, in our setup radiation damage is extremely unlikely (see note at the beginning of the “Response to Referees” and new Supplementary Note 3).

From the flight distance of each particle, it seems that the two XFEL pulse irradiation points are sufficiently separated, but did you see any local damage to the side chains of the refined structure of the hemoglobin?

No we did not, see also response to referee 2 concerning this point.

In order to clarify the distinction between shock wave damage, which is the subject of this paper, and radiation damage that has been known so far, I think it is better to briefly describe the radiation damage of the results of this structural analysis.

There are no radiation damage results in this manuscript. But given the comments also from the other reviewers (see note at the beginning of the “Response to Referees”) we have added a new Supplementary Note 3 to the supplement.

Reviewer #2 (Remarks to the Author):

In this work, Grünbein et al. evaluated the influence of so-called shockwave during serial femtosecond crystallography (SFX) measurements on protein crystal samples using the properties of LCLS with a view to developing a measurement method in EuXFEL. Although disturbance of the sample stream from the injector by shockwave has conventionally been considered only an obstacle in SFX measurements, the authors precisely analyzed the properties of shockwave.

In doing so, the authors not only contributed to the establishment of new techniques for SFX measurement methods, but also presented fundamental findings for the future development of novel analytical methods for understanding biochemistry, such as protein folding.

I would like to request that the authors answer the following questions/comments.

About Figure 1, the position of the GDVN nozzle should be clearly indicated so that the direction of the sample flow can be recognized at a glance, because in the current Fig, non-specialists may think that the sample flows from bottom to top.

How about also describing the length between the sample flow and the iron filter?

We changed the figure. Concerning the iron filter (10 x 10 cm), it was placed 6 cm behind the jet before the CSPAD detector.

The overall setup of the instrumentation described in the Methods section appeared to be elaborate. However, since the layout of the instruments is difficult to understand in a text description alone, why don't authors provide supplementary photos of the actual instrumentation so that it can be easily reproduced/verified by other researchers?

We thank the referee for the positive feedback of our description of the experimental setup. We included a photo of the arrangement inside the experimental chamber (see below). The picture does not include the injector nozzle and the surrounding shroud, which would block the view on other components in the chamber. We instead indicated the position of the jet.

We do not think that this would be of immediate help to the reader, because the experimental setup inside the chamber consists of many items, most of them standard in SFX, such that it is difficult to capture everything in one picture given the spatial constraints of the vacuum chamber. Moreover, important features of the experiment are of different dimensions (micron-sized jet and X-ray interaction regions vs. millimeter- and centimeter-sized experimental equipment for imaging and X-ray detection) such that a photograph cannot capture the situation. We therefore highlighted the non-standard components (e.g., the jet imaging setup, the iron filter in front of the detector, the relative arrangement of pump, probe and jet) in words in the descriptions of the methods.

What is the focused beam size (400 nm?) of the XFEL pump/probe? It should be clearly stated in the Methods section.

We thank the referee for noticing this omission. The beam size of both the pump and the probe beam was around $1.5 \times 1.5 \mu\text{m}^2$ (FWHM), we added this to the Methods section.

Why is the resolution of haemoglobin structure so poor in this study, even in single-pulse scheme data? Simply because of low pulse energy? I had imagined that haemoglobin crystals diffract to high resolutions more than 2 Å.

The referee is right that many hemoglobin crystal forms diffract to high resolution. When we grow large macroscopic Hb crystals using similar crystallization conditions, we typically get 1.6-2 Å resolution of cryocooled crystals at the Swiss Light Source (see also Safo and Abraham, Biochemistry 44: 8347-8359 2005, RR2 form, resolution 2.18 Å), only very occasionally better. We think it is the high plasticity of this crystal form that limits the resolution. Obviously, having a weak incident X-ray intensity does not help. However, the microcrystals are in general not very “happy”, they diffract much worse than large crystals at a synchrotron. We collected a SFX dataset using a fixed target setup at SACLA of these crystals, the resolution was 2.2 Å (Doak et al, Acta Cryst D, 2018, <https://doi.org/10.1107/S2059798318011634>).

To allow the reader to visually evaluate the quality of the solved structure, the authors must include electron density maps as supplementary Figures. In particular, I would like to see the electron density around the haem cofactor.

We thank the referee for his/her suggestion and added such a figure to the Methods section (Supplementary Figure 14)

I am concerned that the crystals/proteins could have been damaged not by shockwave, but by pump XFEL's scattered light that reached the probe position. Is this possibility ruled out? (Even if it's due to scattered light, I think it's a worthwhile study.)

The referee is correct in his concerns, we were too. However, if one goes through the numbers, it is extremely unlikely (see note at the beginning of the “Response to Referees” and new Supplementary Note 3).

I think the degree of conformational doming of haem (a curvature of the haem plane) varies with photolysis or metal redox state within 1 ns period (e.g., Levantino et al. Struct. Dyn. 2015). By comparing the conformations of haem cofactors between single-pulse data and pump-probe data, it may be possible to see if photolysis/photoreduction has occurred (although the resolution for that may be insufficient). Alternatively, the authors may be able to test whether photoreduction occurred using other photosensitive samples instead of hemoglobin crystals in the future work.

X-ray photoreduction is the reduction of redox sensitive groups or molecules (metal centers, flavins, S-S bonds, ...) by photoelectrons, generated by the photoelectric effect. It is thus a manifestation of radiation damage. As described in detail above, radiation damage is highly unlikely in our experiment. Nevertheless, since the referee asked, judging radiation damage in carbonmonoxy hemoglobin is not straight forward. No changes occur due to X-ray photoreduction, the Fe is already reduced and positioned in the heme plane. The bound CO does not dissociate. (Levantino et al performed an optical pump X-ray probe experiment (similar to Barends et al Science 2015 on Mb.CO) resulting in optical photolysis of the Fe-CO bond, and thus also changes in heme doming, but this is not related to radiation damage.) The referee may be aware of recent publication describing ultrafast X-ray induced radiation damage in SFX experiments, Nass, Gorel et al, Nature Commun. (2020) 11:1814 <https://doi.org/10.1038/s41467-020-15610-4>). X-ray induced changes are readily observed in S-S bridges. However, hemoglobin does not have any S-S bridges.

Nonetheless, we are confident that radiation damage is extremely unlikely in view of our experimental setup (see note at the beginning of the “Response to Referees” and new Supplementary Note 3).given the r set up of our experiment aiming at avoiding exactly that.

Minor points:

Page 4, line 90

“Such “packing defects” and the large solvent channels in protein crystals render proteins and their crystals sensitive to applied pressure.”

Please cite references if available.

We added appropriate references to the manuscript, thank you.

Page 12, line 346

“such that only diffraction patterns generated by the pump pulse were recorded.”

the pump pulse -> the probe pulse

Indeed, this was wrong. Changed, thank you.

Page 14, line 395

“the distance between thee detector and the XFEL”

thee -> the

Done.

In Table 1, R_{work}/R_{free} for pump probe in Automatically determined high resolution limits 0.18841 (0.27316) -> 0.18841 / 0.27316

Done.

Eiichi MIZOHATA

Reviewer #3 (Remarks to the Author):

Under a specific set of conditions, the experiments reported by Grünbein et al are intended to assess the impact of shock waves traveling up a liquid column that were created by an XFEL X-ray pulse. In these results, the shock wave is assumed but not visualized as limited by the small diameter of the liquid jet. The timing delay and offset between the pump and probe are 122.5 ns and $\sim 5 \mu\text{m}$ upstream in the liquid jet, respectively. These conditions probe an analogous region of the jet anticipated by the 4.5 MHz Eu.XFEL conditions, with the caveat that the delay between the pump and probe is shorter by about half. The shock wave(s) travels at supersonic speed and will have passed by this region of the jet before the region is probed with the second X-ray pulse.

This is the first of this type of experiment. These are very interesting results and should be published, after considering the comments below.

The temporal response to the pressure jump is probed with only one time point (~ 122 ns) and with only one type of sample (haemoglobin CO complex, Hb.CO) which does demonstrate changes in the atomic models fit to the pump-probe SFX diffraction data. The authors should include a statement on the estimated errors of the atomic models as these are likely to be similar to the magnitude of the changes observed in these pump-probe results.

The referee is correct, this can be a major issue when using traditional refinement procedures at medium to high resolution. Therefore we have used a bootstrapping approach to estimate the accuracy of our refinement results. We also point out that the coordinate error is smaller for correlated displacements involving many atoms such as helices. The referee may have missed this, it is described in detail in the methods section which is also referred to in the main text.

In contrast, and as briefly stated by the authors (lines 242 - 250), experiments at the Eu.XFEL reported by Yefanov et al ((2019) Struct Dyn 6, 064702) indicate that with a 100 m/s liquid jet and a 1.1 MHz pulse frequency, there is no observable difference in atomic models of hen egg while lysozyme refined against SFX datasets derived from each of the sequential pulses within the 1.1 MHz pulse train. The differences between experimental conditions between the two reports are rather significant (please see the attached image -- Illustrated to approximate scale) and should be discussed further.

We have discussed the previous experiments performed at EuXFEL (Gruenbein et al, Wiedorn et al, Yefanov et al) in the section “Implications for data collection at the EuXFEL”. Yefanov et al, and to some extent also Wiedorn et al, use 100 m/s jets. Gruenbein et al and Wiedorn et al have used a much larger X-ray focus than Yefanov et al. The referee misses one important point: In contrast to all previous published experiments at EuXFEL, including Yefanov et al, by using fs jet imaging, we here ensure that a given sample segment had indeed experienced a shock wave. The three EuXFEL publications simply average all diffraction patterns collected at a given position in the pulse train independently of whether the previous XFEL pulse actually intersected the jet, thus launching a shock wave.

More importantly, as mentioned by the referee, the different combinations of jet speed, X-ray repetition rate and jet diameter in the various experiments lead to a different pressure experienced by the probed crystals. From the referee’s comment we conclude that our comparisons were not explicit enough. We have therefore edited this section to make both points more explicit. We also estimated (see Supplementary Note 5) that the shock pressure in the Yefanov experiment is at least one order of magnitude lower than the pressure in our experiment; therefore, it is not surprising that we observed shock-induced changes and Yefanov et al. did not.

Indeed, Grünbein et al state that their use of a 50 m/s jet speed is the “minimum allowable” for SFX measurements collected with MHz pulse frequencies at the European XFEL. In fact, their data presented herein suggests that SFX data collection at MHz XFEL sources should probably be carried out with jets faster than 50 m/s.

We thank the referee for catching this mistake. The referee is completely right, the conclusion of the experiments described in this manuscript is that jet of 50 m/s can most likely not be used for collecting undamaged data at MHz X-ray repetition rate. The ‘minimum allowable’ speed was an assumption prior to our experiments, which we then tested in our experiments. We clarified in the text that using jets of 50 m/s was based on an assumption, and explicitly restated in the section “Implications for data collection at EuXFEL” that jets faster than 50 m/s are required to collect undamaged data at 4.5 MHz repetition rate.

The pump-probe SFX diffraction data was obtained from rod-shaped Hb.CO crystals that were of similar size to the jet diameter and the offset of the two X-ray pulses. This reviewer looked for, but could not find, the size of either the pump or probe X-ray beams; both of these must be included.

We thank the referee for noticing this omission. The beam size of both the pump and the probe beam was around $1.5 \times 1.5 \mu\text{m}^2$ (FWHM). We added this information to the Methods section.

The crystal density in the slurry was not listed and should be given.

The density was about 15 % (v/v) settled crystalline material. We added this information to the Methods section.

There was a statement (supplementary note 2; “flow alignment of the Hb.CO microcrystals in the liquid jet”) that there may be a preponderance of the long axis of the rods aligned with the jet stream. Thus, it is also possible that a 10 μm long crystal rod could be hit by the weaker pump pulse and then again by the stronger probe pulse.

The concerns of referee are in principle correct. However, he/she likely overlooked the fact that Double hits are not possible because the crystals also move during the time delay between pump and probe pulses, see section at the beginning of the Response to Referees. Some flow alignment was apparent when displaying powder patterns calculated from crystal hits during data collection. We have plotted the orientation matrices (missetting angles) of the crystal hits, see new Supplementary Figures 9-11.

If so, then these crystals might exhibit reduced diffraction quality due to more traditional radiation damage mechanisms, rather than the presumed pressure wave(s).

Since all three referees raised this concern we address it at the beginning of the “Response to Referees” and new Supplementary Note 3. The low degree of flow alignment combined with relative dimensions of crystal size, X-ray beam size, distance between pump and probe and distance translated by crystals between the two pulses leads to a negligible probability that the reduction in diffraction quality is caused by such double hits. We also discuss in the Supplementary Note 3 why radiation damage due to radicals or electrons is extremely unlikely in our experiment.

In conclusion the observed deterioration of diffraction quality cannot due to X-ray radiation damage but must be caused by shock damage, an effect that can overcome longer distances.

It is unlikely that the observed and reported overall statistical differences will derive from a relatively small number of examples that may fall into this particular scenario. Without more information on the crystal density and estimate of the propensity to flow align, one cannot estimate the potential impact to the experiment.

The referee is correct that we forgot to indicate the crystal density (15 % (v/v) settled microcrystalline material), this information is now added to the methods section.

However, we emphasize that the probability of the aforementioned “double hits” affecting the resulting data would be independent of the crystal density. Even if it were possible to obtain double hits (which in our case is highly unlikely, see section at the beginning of “Response to Referees” and Supplementary Note 3) the probability of a double hit (# double hits/# X-ray shots) increases with crystal density, but so does the probability of any kind of hit (# hits / # X-ray shots). Thus, the fraction of double hits relative to all hits remains constant for any crystal density and solely depends on the relative geometry of the crystal, pump and probe location and

dimension as well as jet speed and time delay between pump and probe (see Supplementary Note 3).

The use of the two-bunch mode at the LCLS does impact the experiment since the pump pulse had significantly lower photon pulse energy and impacting the resulting pressure waves compared to the probe pulse. The stated ~ 0.03 mJ (line 470) on average and up to ~ 0.1 mJ per pump pulse is significantly lower than the ~ 2 mJ per pulse in earlier studies (especially Stan et al (2016) Nat Phys 12, 966-971) that characterized the shock waves in larger diameter jets.

We had addressed this point in the section “Implications for data collection at the EuXFEL”. The referee might have missed it.

The published “record” for the lowest-energy-XFEL-pulse that generates a measurable shock wave is Stan et al., Appl. Sci 10: 1497 (2020) <https://doi.org/10.3390/app10041497>): 7 μ J at sample at 10 keV (lower photoabsorption cross section than at 7 keV) see Fig 3, Fig 5b. This is a very good reason to expect shocks from 0.03 mJ, despite the fact that we cannot image them. Shocks can be generated by much weaker pulses than the ones used in the earliest studies.

We also estimated (see Supplementary Note 5) the shock pressure in our experiment and compared it to the Yefanov experiment.

However, the authors present femtosecond snapshot imaging results that indicate that the pump pulse does cut the jet and by extension also likely induces shock waves along the liquid column. It is not clear how these different experimental conditions translate into pressure waves impacting crystals within the slurry of the liquid jet.

We’re not sure what point the referee wants to raise here. We have shown that the experimental conditions are sufficient to generate explosions in the jet, and by design of the experiment, radiation damage is highly unlikely (see above). Despite the low pulse energies, the focused beam induces explosions and launches shock waves, just of lower magnitude. We address this in the section “Implications for data collection at the EuXFEL”, and we included an estimate of the shock pressures in the revised manuscript (see above).

Lines 158-159 and Table 1

The authors state, "We collected 43,003 and 25,742 hits, respectively, for the pump-probe and single-pulse setups." Although, “hits” should probably be more explicitly defined as crystal lattices, the implication is that they have rather robust and complete datasets. A smaller number of these passed the various filter analysis tests (14,434 and 24,083). Thus, the pump-probe experimental regime resulted in retention of only $\sim 33\%$ of the lattices compared to $\sim 94\%$ of the single-pulse experiments.

As we state in the Methods section, any detector image in which at least 10 peaks were detected by CASS is considered a hit. We did not see the need to distinguish between scattering patterns

(no crystal), diffraction patterns (crystalline material) and crystal hits (diffraction pattern with > 10 Bragg reflections). Technically a diffraction pattern consistent with a powder pattern would not be classified a crystal hit by CASS (we would also not analyze it in this experiment) but it should be classified as “crystal lattice”(and could in principle be analyzed). For this and other reasons we prefer not to use the term “crystal lattice” for what we label “hit”. We now specify “hit” as “crystal hit” in the revised version of the manuscript.

For all of pump-probe crystal hits, the jet image and the X-ray diode data is analyzed to ensure using only data with clean (non-noisy) diode signal, with a pump pulse energy above the Fe K edge and a jet that allows shock wave propagation. In particular the jet imaging filtering step of the pump-probe data set is rather restrictive (see below).

Then in Table 1, there are only 3500 “diffraction images used” which again should probably be crystal lattices and is only about 8% of original pump-probe “hits”. The data only extends to modest resolution (2.54 – 2.77 Å) even with liberal cut-off of parameters.

Although the authors discuss a desire to compare datasets from as similar conditions as possible, it is not at all clear where in the process such a large fraction of the lattices get rejected?

We want to compare non-shocked and shocked crystal diffraction images. This means we need to be sure that the latter did indeed experience a shock wave launched by the previous pulse. This may sound trivial (just take the data collected with the second X-ray pulse, see EuXFEL publications by Gruenbein et al, Wiedorn et al, Yefanov et al) but it is not. Due to jet (and in principle also X-ray pointing) instabilities it is quite possible that the first pump pulse missed the jet, but the second did not. In this case, the probe beam intersected a non-shocked crystal. We need to avoid this possibility since we are interested in shocked data only and not in maximizing the number of crystalline hits for structure determination.

Data is rejected on the basis of the X-ray diode data and the jet image as explained above, not on a crystallographic analysis of the CSPAD diffraction data. Particularly the filtering based on the jet images leads to a significant fraction of crystal hits being rejected: We employ stringent filtering conditions to ensure that crystal hits passing the filtering have indeed experienced a shock wave (see our Supplemental Information). Less stringent filtering conditions may wash out the shock wave effect or render it undetectable. Since by definition there is no shock wave in the single pulse data set, these hits are not passed through the jet image filter, leading to much less hits being rejected for the single-pulse data set.

We agree with the referee, though, that the number of hits and indexed hits as well as the hit and indexing rate before and after filtering should be mentioned more clearly in the paper. Below we include the respective numbers that we also included into the manuscript.

	No filtering		After filtering	
	Pump-probe	Single-pulse	Pump-probe	Single-pulse
# hits	43 003	25 742	14 434	24 083
Hit rate	13 %	19 %	*	*
# indexed hits	12 280	5 958	3 531	5 541

Indexing rate	29 %	23 %	24 %	23 %
-------------	-------------	-------------	-------------

**hit rate not defined on the filtered data set*

Line 118 and line 717 (Ref 28).

Reference 28 is very a hard to find; please add another more accessible reference to assist the readers.

Unfortunately this work has not been published anywhere else. We added the direct link to help with the search

<http://accelconf.web.cern.ch/fel2017/doi/JACoW-FEL2017-TUP023.html>

Line 341:

Please indicate the approximate size of the X-ray beams at both the pump and probe locations.

Done, see above

Line 542 “ β 1core, ...).”

The use of three dots makes it appear that this legend has not been completed?

We exchanged the three dots to “etc”.

REVIEWER COMMENTS

Reviewer #1 (Remarks to the Author):

The authors have explained their point sufficiently. No further comment from me.

Reviewer #2 (Remarks to the Author):

The manuscript has been much improved. I think it is ready for publication.

Reviewer #3 (Remarks to the Author):

The rebuttal letter is complete and the revised manuscript address all of my concerns; thank you, the manuscript is nicely improved. The spreadsheet for estimating the "Probability of Pump-Probe Dual Hit vs. Crystal Length for various Jet Speeds" is useful. The new supplemental notes also address the potential for double hits and radiation damage.

There are at least two references (e.g. 1 and 4) missing the "et al" wherein a large number of co-authors are included on the publications.

Following the links in reference 31 produced errors and thus the article is still hard to find by this reviewer.

31 Decker, F.-J., Bane, K.L.F., Colocho, W.S., Lutman, A.A. & Sheppard, J.C. in International Free Electron Laser Conference. (eds Kip Bishofberger, Bruce Carlsten, & Volker RW Schaa)
<http://accelconf.web.cern.ch/fel2017/doi/JACoW-FEL2017-TUP2023.html> (JACoW,
<https://doi.org/10.18429/JACoW-FEL2017-TUP023>, 2018).

"HTTP Error 404.0 - Not Found

The resource you are looking for has been removed, had its name changed, or is temporarily unavailable."

and

"DOI Not Found

10.18429/JACoW-FEL2017-TUP023"

Referees 1 and 2 had no further comments.

Referee 3 remarked:

There are at least two references (e.g. 1 and 4) missing the “et al” wherein a large number of co-authors are included on the publications.

We fixed the formatting. We thank the referee to find this mistake

Following the links in reference 31 produced errors and thus the article is still hard to find by this reviewer.

We could not reproduce the problem. Two different people tried and had not issues locating the reference.